# Compatible interaction of *Brachypodium distachyon* and endophytic fungus *Microdochium bolleyi*

**Pavel Matušinsky**[1,2]*, **Božena Sedláková**[1], **Dominik Bleša**[2,3]

**1** Department of Botany, Faculty of Science, Palacký University in Olomouc, Olomouc, Czech Republic, **2** Department of Plant Pathology, Agrotest Fyto, Ltd, Kroměříž, Czech Republic, **3** Department of Experimental Biology, Faculty of Science, Masaryk University, Brno, Czech Republic

* pavel.matusinsky@upol.cz

**Data Availability Statement:** All relevant data are within the paper and its Supporting information files.

**Funding:** Authors of this study were supported by the Ministry of Agriculture of the Czech Republic,

## Abstract

*Brachypodium distachyon* is a useful model organism for studying interaction of cereals with phytopathogenic fungi. The present study tested the possibility of a compatible interaction of *B. distachyon* with the endophytic fungus *Microdochium bolleyi* originated from wheat roots. There was evaluated the effect of this endophytic fungus on the intensity of the attack by pathogen *Fusarium culmorum* in *B. distachyon* and wheat, and also changes in expression of genes (in *B. distachyon*: *BdChitinase1*, *BdPR1-5*, *BdLOX3*, *BdPAL*, *BdEIN3*, and *BdAOS;* and in wheat: *TaB2H2(chitinase)*, *TaPR1.1*, *TaLOX*, *TaPAL*, *TaEIN2*, and *TaAOS*) involved in defence against pathogens. Using light microscopy and newly developed specific primers was found to be root colonization of *B. distachyon* by the endophyte *M. bolleyi*. *B. distachyon* plants, as well as wheat inoculated with *M. bolleyi* showed significantly weaker symptoms on leaves from infection by fungus *F. culmorum* than did plants without the endophyte. Expression of genes *BdPR1-5*, *BdChitinase1*, and *BdLOX3* in *B. distachyon* and of *TaPR1.1* and *TaB2H2* in wheat was upregulated after infection with *F. culmorum*. *M. bolleyi*-mediated resistance in *B. distachyon* was independent of the expression of the most tested genes. Taken together, the results of the present study show that *B. distachyon* can be used as a model host system for endophytic fungus *M. bolleyi*.

## Introduction

*Brachypodium distachyon* (L.) P. Beauv (Bd) was proposed two decades ago as a model system for cereals [1], and research since that time has confirmed this to be a proper choice. Like *Arabidopsis*, *Brachypodium* has small stature, short generation time, the ability to self-pollinate, and it is easily grown under simple conditions [1]. In addition, Bd has one of the smallest genomes found in grasses [2], comprising just 5 chromosomes spanning 272 Mbp and within which about 25,000 protein-coding sequences are predicted [3]. Host-pathogen interactions between *B. distachyon* and plant pathogens have previously been described for a number of important cereal diseases [4]. Phytopathogenic fungi with compatible interaction have been

projects numbers QK1910197, QK21010064, MZE-RO1118 and Internal Grant of Palacky University project number IGA_PrF_2022_002. The funders had no role in study design, data collection and analysis, decision to publish, or preparation of the manuscript.

**Competing interests:** The authors have declared that no competing interests exist.

described between *B. distachyon* and *Rhizoctonia solani*, *Claviceps purpurea*, *Ramularia collo-cygni*, *Oculimacula* spp., *Magnaporthe grisea*, *Cochliobolus sativus*, *Gaeumannomyces graminis*, *Pyrenophora teres*, *Fusarium* spp., *Stagonospora nodorum*, and *Colletotrichum cereale* [5–10]. Multiple studies have confirmed interactions among Bd and such rust pathogens as *Puccinia graminis*, *Puccinia triticina*, *Puccinia hordei*, and *Puccinia striiformis* [11–13]. Bd is also susceptible to some oomycota [5], bacteria [4], and viruses [14, 15] previously described as cereal pathogens.

A study by Peraldi *et al*. confirmed positive interaction between the species most prevalently causing Fusarium head blight (FHB) and Bd [9]. As documented in that study, *Fusarium graminearum* and *Fusarium culmorum* successfully infected tissue of Bd. *F. culmorum* is a soil-, air- and seed-borne fungal pathogen of small-grain cereals causing foot and root rot, fusarium seedling blight, and especially FHB, a disease leading to decreased yield and mycotoxin contamination of grain [16]. *Fusarium culmorum* is observed in colder regions of Europe, America, Australia, Asia, and North Africa [17–20] and is regarded globally as one of the main pathogens of cereals [21, 22]. Because no fully resistant cultivars exist, control of diseases caused by *Fusarium* spp. must be achieved by such agricultural management practices as crop rotation and postharvest debris removal to diminish inoculum pressure [23]. Fungicide treatments at anthesis can reduce disease levels by only 15–30% [24], and, together with rising environmental concerns, new disease control alternatives, including biological controls, need to be developed. Several organisms have been tested both *in vitro* and in plant assays for their efficacy to control FHB in wheat [25–27]. Endophytic biological control agents may offer such potential [28].

*Microdochium. bolleyi* (Sprague) de Hoog and Hermanides-Nijhof, *(Ascomycota, Xylariales)* (Mb) is a fungus endophytically growing within plant roots, especially in cereals and other graminaceous species [29]. It has been documented on wheat [30], barley [31, 32], and other plants, such as *Agrostis stolonifera* L. [33], *Agrostis palustris* [34], and *Phragmites australis* [35]. Mb is characterized as a dark septate endophyte due to its melanised cell walls and intra- and intercellular growth within the roots of healthy plants [36]. In culture, Mb produces one-celled, crescent-shaped conidia and dark brown hyphae, and it may release an orange pigment [37]. Mb has been shown to exhibit suppression of various plant pathogens of cereals, such as *Oculimacula yallundae* [30], *G. graminis* var. *tritici* [32, 38], *Septoria nodorum* [39], and *Bipolaris sorokiniana* [40]. Mb inhibited the growth of *F. culmorum* by 24.5–33% in dual culture *in vitro*, and symptoms on detached spikelets caused by *F. graminearum* were lower to 54% in treatment by Mb [41].

The present research aims to investigate the interaction between *M. bolleyi* and *B. distachyon* and evaluate Bd as a model for examining potential of endophyte-mediated resistance to *F. culmorum* (Fc). A second aim was to develop a method for molecular identifying Mb in plant tissues.

## Materials and methods

### Biological materials

Seeds of the *B. distachyon* line Bd21 were obtained from the Joint Genome Institute (https://jgi.doe.gov). All plants were maintained in a cooled greenhouse (20/18˚C, day/night) within pots 8 cm in diameter filled with a 50:50 planting substrate–sand (vol/vol) mixture, the substrate being FLORCOM SV (BB Com, Letohrad, Czech Republic). Ten seeds were placed into each pot. After germination, the number of plants in every pot was reduced to six.

The Mb isolates originated from wheat roots. There were used six isolates (UPOC-FUN-253–258) from the Collection of Phytopathogenic Microorganisms UPOC (Czech Republic).

All isolates come from the Czech Republic and were collected during the years 2018 and 2019. Inocula of Mb isolates were cultivated on millet grains for three weeks at 20°C in the dark. Before cultivation, 200 g of millet seeds and 50 ml of distilled water in each plastic bag were steam sterilized twice at 120°C for 20 min. Each isolate was cultivated separately, and then, prior to use, all six isolates were mixed all together from equal parts. The Fc isolate 19FcBd was obtained from symptomatic leaves of Bd in a greenhouse during 2019. The isolate was checked for colony morphology, conidial morphology, and sequencing of the internal transcribed spacer (ITS), large subunit (LSU), and elongation factor (EF-1). Sequences were checked using the online Basic Local Alignment Search Tool (BLAST). Best hits were examined to attribute species names (≥97% of sequence similarities). Species-specific primers OPT18F/R [42] were also used to confirm Fc determination. Cultures were maintained in darkness at 18 ± 2 °C on potato dextrose agar (PDA) plates and transferred regularly to fresh medium. The Fc inoculum was cultured on PDA plates under UV-B light at 18 °C for 2 weeks. Macroconidia were obtained by scraping the agar surface with a sterile spatula and transferring the conidia to sterile distilled water. A final solution of Fc macroconidia was prepared at a concentration of $5 \times 10^5$ conidia $mL^{-1}$.

Isolates of other fungi used for testing of diagnostic primers were obtained from four different collections of microorganisms (Table 1).

## Inoculations

Plants were inoculated with Mb during sowing. Millet containing Mb (2.5 g per pot) was spread evenly directly on the seed placed on the substrate. Treatments inoculated with Mb are indicated within this article as Mb1. Only sterile millet without endophyte was added to the control treatments. Treatments non-inoculated with Mb are indicated within this article as Mb0. Seed and inoculum were overlaid with a 0.5 cm layer of the substrate.

In the phase of the second offshoot in Bd and the third leaf of wheat was carried out infection using Fc on the second upper leaf, adapting and modifying the method of Peraldi et al. [9]. Treatments infected with Fc are indicated within this article as F1. Leaves were wounded in two positions by gentle compression with a Pasteur pipette on the adaxial surface [9]. A droplet (5 μL) of conidial suspension (containing $5 \times 10^5$ conidia $mL^{-1}$), amended with 0.05% Tween 20, was deposited onto each wound site. Mock inoculation was performed similarly using sterile distilled water with 0.05% Tween 20 (5 μL). Treatments non-infected with Fc are indicated within this article as F0. All plants were placed under a plastic cover to increase relative humidity until 2 days post inoculation (dpi), at which time the covers were removed.

## Sampling and assessment

Disease symptoms on the leaves were recorded at 8 dpi in Bd and wheat. Leaves for light microscopy were collected at 8 dpi and fixed in 70% ethanol. The percentage of necrotic tissue was evaluated on the adaxial surface of infected leaves (S1 Fig). Leaves for RNA extraction were sampled at 1, 2, and 8 dpi and immediately transferred to liquid nitrogen. Subsequently, were transferred to a freezer and held at −80 °C.

For the purpose of evaluating endophyte colonization by DNA extraction, roots were collected 90 days after sowing, surface sterilized for 3 min in 1% NaOCl and thoroughly rinsed in sterilized distilled water to remove all superficial hyphae and then dried. Roots for the purpose of evaluating endophyte colonization by light microscopy were collected 90 days after sowing, then fixed in 70% ethanol. Leaves and roots fixed in 70% ethanol were subsequently cleared in 2.5% KOH for 3 days, acidified in 1% HCl, then stained with 0.05% aniline blue in lactoglycerol [43]. Colonization of roots by the endophytic fungus was assessed by microscopic

**Table 1. Fungal species used in the analysis with MbPOLIIF/R primers designed in the current study.** A positive reaction result distinguished by a visible band on the gel is indicated by a + sign, a negative response by a –sign. To confirm the presence of DNA, a control polymerase chain reaction (PCR) reaction with universal primers ITS1/ITS4 was performed on all samples. Reactions were always repeated three times.

| Species | Source | Code | Host | ITS1/4 | MbPOLIIF/R |
|---------|--------|------|------|--------|------------|
| *Microdochium bolleyi* | UPOC | UPOC-FUN-253 | *Tritium aestivum* | +/+/+ | +/+/+ |
| *Microdochium bolleyi* | UPOC | UPOC-FUN-254 | *Tritium aestivum* | +/+/+ | +/+/+ |
| *Microdochium bolleyi* | UPOC | UPOC-FUN-255 | *Tritium aestivum* | +/+/+ | +/+/+ |
| *Microdochium bolleyi* | UPOC | UPOC-FUN-256 | *Tritium aestivum* | +/+/+ | +/+/+ |
| *Microdochium bolleyi* | UPOC | UPOC-FUN-257 | *Tritium aestivum* | +/+/+ | +/+/+ |
| *Microdochium bolleyi* | UPOC | UPOC-FUN-258 | *Tritium aestivum* | +/+/+ | +/+/+ |
| *Microdochium nivale* | AGT | 13M30 | *Tritium aestivum* | +/+/+ | -/-/- |
| *Microdochium nivale* | AGT | 17M323 | *Tritium aestivum* | +/+/+ | -/-/- |
| *Microdochium nivale* | AGT | 13M205 | *Tritium aestivum* | +/+/+ | -/-/- |
| *Microdochium majus* | AGT | 13M195 | *Tritium aestivum* | +/+/+ | -/-/- |
| *Microdochium majus* | AGT | 14M71 | *Tritium aestivum* | +/+/+ | -/-/- |
| *Microdochium majus* | AGT | 17M271 | *Tritium aestivum* | +/+/+ | -/-/- |
| *Cochliobolus sativus* | AGT | 07CS4.3 | *Hordeum vulgare* | +/+/+ | -/-/- |
| *Ramularia collo-cygni* | AGT | 20CZR19 | *Hordeum vulgare* | +/+/+ | -/-/- |
| *Oculimacula yallundae* | AGT | 15OY119 | *Tritium aestivum* | +/+/+ | -/-/- |
| *Oculimacula acuformis* | AGT | 15OA103 | *Tritium aestivum* | +/+/+ | -/-/- |
| *Rhizoctonia cerealis* | AGT | 20CC88 | *Tritium aestivum* | +/+/+ | -/-/- |
| *Gaeumannomyces graminis* var. *tritici* | CCM | F-575 | *Tritium aestivum* | +/+/+ | -/-/- |
| *Rhynchosporium secalis* | AGT | 18RhS04 | *Hordeum vulgare* | +/+/+ | -/-/- |
| *Pyrenophora teres* | AGT | 17PTT52 | *Hordeum vulgare* | +/+/+ | -/-/- |
| *Pyrenophora maculata* | AGT | 14PTM01 | *Hordeum vulgare* | +/+/+ | -/-/- |
| *Pyrenophora tritici-repentis* | AGT | 19DTR6 | *Tritium aestivum* | +/+/+ | -/-/- |
| *Tilletia tritici* | AGT | 06TCAR33 | *Tritium aestivum* | +/+/+ | -/-/- |
| *Tilletia controversa* | AGT | 06TCO02 | *Tritium aestivum* | +/+/+ | -/-/- |
| *Fusarium graminearum* | AGT | 20FG01 | *Tritium aestivum* | +/+/+ | -/-/- |
| *Fusarium culmorum* | AGT | 19FcBd | *B. distachyon* | +/+/+ | -/-/- |
| *Fusarium avenaceum* | CPPF | CPPF-161 | *Tritium aestivum* | +/+/+ | -/-/- |
| *Fusarium poae* | CPPF | CPPF-51 | *Tritium aestivum* | +/+/+ | -/-/- |
| *Fusarium langsethiae* | AGT | 12FL4.00 | *Avena sativa* | +/+/+ | -/-/- |
| *Fusarium sporotrichioides* | CPPF | CPPF-146 | *Tritium aestivum* | +/+/+ | -/-/- |
| *Fusarium tricinctum* | CPPF | CPPF-254 | *Tritium aestivum* | +/+/+ | -/-/- |
| *Fusarium oxysporum* | AGT | 19FOX06 | *Zea mays* | +/+/+ | -/-/- |
| *Zymoseptoria tritici* | AGT | ST-KM_B | *Tritium aestivum* | +/+/+ | -/-/- |
| *Penicillium sp.* | AGT | 20PEN_sp. | *Hordeum vulgare* | +/+/+ | -/-/- |

UPOC–Collection of Phytopathogenic Microorganisms, Czech Republic; AGT–Agrotest Fyto, Ltd, Czech Republic; CCM–Czech Collection of Microorganisms, Masaryk University, Faculty of Sciences, Czech Republic; CPPF—Collection of Phytopathogenic Fungi at Crop Research Institute Prague, Czech Republic.

examination (200× magnifications). Results were evaluated as positive when visible presence of Mb chlamydospores was observed.

## DNA isolation, primers design, and PCR

Fungal mycelia (approximately 50–100 mg of biomass) of all tested species and strains (Table 1) were harvested from Petri dishes, ground to a fine powder in a cooled mortar using liquid nitrogen, then homogenized. Total genomic DNA was extracted using the DNeasy Plant

Mini Kit (Qiagen, Germany). Plant material (roots of Bd and wheat, approximately 50 mg of dried biomass per sample) was also ground to powder and extracted as described above. DNA concentration was measured using Qubit fluorometric quantification (ThermoFisher Scientific, Waltham, MA, USA) and DNA was diluted to concentration 5 ng μL$^{-1}$.

Based upon sequences ITS (KP859018), LSU (large subunit of ribosomal gene; KP858954), Tub2 (β-tubulin gene; KP859081), and *RPB2* gene (RNA 40 polymerase II second-largest subunit (KP859127)) [44] selected from the GenBank database, four primer pairs (MbITSF/R, MbLSUF/R, MbBETF/R, and MbPOLIIF/R) were designed using Primer3Plus software [45]. PCR reactions were carried out in 20 μL volumes containing 10 ng of DNA from fungal and/ or plant material. The reaction mixture consisted of 0.2 mM of dNTP, 0.2 μM each of forward and reverse oligonucleotide primer (the best final primers MbPOLIIF/R were selected from a preliminary screen of the designed primers (see S1 Table)) and 1 U of Taq polymerase. Reaction buffer consisted of 75 mM Tris-HCl, 20 mM (NH$_4$)$_2$SO$_4$, and 2.5 mM MgCl$_2$.

DNA extracted from fungal cultures and from infected and non-infected plant material was amplified using PCR with initial denaturation 94 ˚C (5 min). The temperature cycle (35×) consisted of denaturation (95 ˚C) for 30 s, annealing 66.7 ˚C (20 s), and extension at 72 ˚C (45 s). A final extension step at 72 ˚C for 5 minutes was followed by cooling to 10 ˚C until removing samples. PCR products (10 μL) were analysed using agarose gel electrophoresis.

## Test of primers sensitivity, specificity, and diagnostic potential in plant tissues

The primers were tested on the six Mb isolates as described above. The sensitivity was tested using different amounts of Mb DNA (1.0, 0.1, 0.01, 0.001, and 0.0001 ng.). The designed primer pairs were also tested for their specificity towards the DNA of fungi associated with diseases of wheat and other cereals. Fungal cultures were obtained from four microorganism collections (Table 1) and cultured on PDA. The quality of fungal DNA was checked by ITS1/ITS4 primers [46]. Furthermore, the potential for detection was tested in Bd and wheat roots inoculated with the Mb compared with non-inoculated roots (inoculation and sampling as described above). All reactions were repeated at least three times.

## RNA isolation and quantitative PCR

Leaves were homogenized in a TissueLyser II (Qiagen, Hilden, Germany) for 2 minutes at 27 Hz. Caution was taken during homogenization to avoid sample melting. The homogenized samples were immediately placed into liquid nitrogen. The RNA was isolated using RNeasy Plant Mini Kit (Qiagen) while following the manufacturer's instructions. DNA was removed during the RNA purification using the RNase-Free DNase Set (Qiagen). The isolated RNA was stored at −80 ˚C. cDNA was synthesized using the Transcriptor High Fidelity cDNA Synthesis Kit (Roche Diagnostics, Mannheim, Germany) according to the manufacturer's instructions with 1 μg of total RNA and anchored-oligo (dT) primers. The concentration of cDNA was measured using Qubit (ThermoFisher Scientific) and cDNA was diluted to concentration 5 ng μL$^{-1}$. Expression analysis of the chosen plant defence genes (*BdChitinase1*, *BdPR1-5*, *BdLOX3*, *BdPAL*, *BdEIN3*, *BdAOS*, *TaB2H2 (chitinase)*, *TaPR1.1*, *TaLOX*, *TaPAL*, *TaEIN2*, and *TaAOS*) was performed using the CFX96TM Real-Time PCR Detection System (Bio-Rad, Hercules, CA, USA). The quantitative PCR mix consisted of 1× SYBR Green (Top-Bio, Vestec, Czech Republic), 0.2 μM forward and reverse primers (S1 Table), 10 ng cDNA (2 μL), and water to final volume 15 μL. The reference gene for the wheat was glyceraldehyde-3-phosphate dehydrogenase (GAPDH) according to Travella et al. [47] and Sun et al. [48], and for Bd, it was S-adenosylmethionine decarboxylase gene *BdSamDC* [49]. The control sample consisted

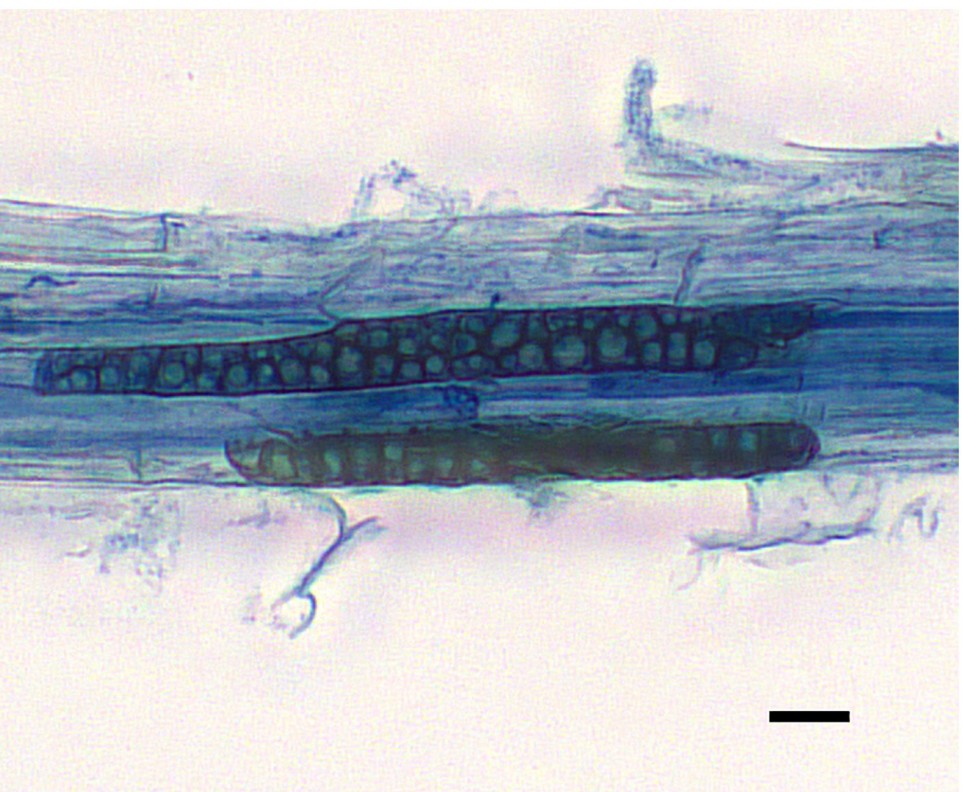

**Fig 1. Chlamydospores of *M. bolleyi* in root of *B. distachyon*.** Scale bar = 10 μm.

of equal amounts of cDNA from all three replications of Bd and wheat plants not inoculated with fungi (with neither Mb nor Fc present). The primers' specificity and presence of primer dimers were verified by melting analysis. The data were analysed using the $2^{-\Delta\Delta Cq}$ method with CFX Manager 3.0 software (Bio-Rad). Three biological as well as three technical replicates were run.

## Statistical analysis

Percentage of necrotic tissue after infection with Fc in the treatments pre-inoculated with the endophyte and non-inoculated were statistically analysed by ANOVA in conjunction with Tukey's post hoc test ($P < 0.05$) using Statistica 12 software. Differences in expression levels of the analysed genes were evaluated by Tukey's test ($P < 0.05$; Maestro software, BioRad).

## Results

### Demonstrated root colonization by Mb of Bd and wheat

Using microscopic and molecular methods, presence of Mb was detected in both Bd and wheat root samples. Using light microscopy and aniline blue staining, chlamydospores were observed in the roots of Bd (Fig 1) and wheat (S2 Fig). No presence of chlamydospores was confirmed in plants without inoculation.

Of the four designed primer pairs, a set MbPOLIIF/R best met the requirements. The primers amplified one sharp band (600 bp) in all six tested isolates of Mb. In sensitivity testing, the

method showed a positive response even at 0.001 ng (1 picogram) of Mb DNA (S3 Fig). In the specificity test, the primers amplified DNA only from Mb and not from any of the other fungal pathogens screened (Table 1). The primer set was therefore used to examine Mb-inoculated and non-inoculated plant material (Bd and wheat). Inoculated plants (Mb1) with chlamydospores present in roots of both Bd and wheat showed positive results after PCR with primers MbPOLIIF/R and, on the contrary, non-inoculated plants (Mb0) without chlamydospores in roots showed negative PCR results (S2 Table). Analysis confirmed the primers to be specific and powerful in detecting Mb within plant tissues. No false positive results and no artefacts were detected after visualization on the electrophoresis. Other primer pairs tested (MbITSF/R, MbLSUF/R, and MbBETF/R) failed in the specificity test and were excluded from other parts of this study.

## Demonstration of successful *Fusarium culmorum* infection

Infection with Fc on leaves of Bd and wheat was successful. This was verified by light microscopy after staining in aniline blue. Germinating macroconidia and hyphae were observed penetrating pipette-damaged tissues (S4 and S5 Figs). Darkening of tissues and incipient necrotization were observed around the penetration of hyphae into the tissues. The intensity of necrotization was assessed visually according to the scheme presented in S1 Fig, and percentage damage was recorded. On plants without Fc infection, darkening of tissues did not occur and areas damaged by pipette remained light coloured. Upon evaluating the data, it was found that plants without endophyte Mb were significantly more severely infected than were plants with Mb (Table 2; S3 and S4 Tables).

## Expression levels of genes involved in plant–pathogen interaction in Bd and wheat

Expression levels of the selected essential plant defence genes (*BdChitinase1*, *BdPR1-5*, *BdLOX3*, *BdPAL*, *BdEIN3*, *BdAOS*, *TaB2H2 (chitinase)*, *TaPR1.1*, *TaLOX*, *TaPAL*, *TaEIN2*, and *TaAOS*) were measured in plants inoculated with endophyte Mb and pathogen Fc. The fold differences (FDs) in their expression levels were first compared among the mock (plants inoculated with neither Mb nor Fc) and other experimental treatments with combinations of single (only Mb or only Fc) and double (both Mb and Fc) inoculations (Fig 2). Significant changes were detected only in some genes (*BdPR1-5*, *BdChitinase1*, *BdLOX3*, *TaPR1.1*, and *TaB2H2*) for treatments infected with pathogenic fungus Fc. Other genes (*BdPAL*, *BdAOS*, *BdEIN3*, *TaLOX*, *TaEIN2*, and *TaAOS*) were not influenced by Fc infection. *BdPR1-5* expression was upregulated by infection of plants with pathogen Fc at 1 dpi and 2 dpi (Fig 2). The average FD across all treatments was 30.94 in the F1 group compared to F0 (Table 3). In the last term, 8 dpi, no significant differences were detected between treatments F0 and F1. In the case of *BdChitinase1*, the expression increase after infection with the Fc pathogen was

**Table 2. Tukey's post hoc test for each species and treatment examining damage to leaves of *B. distachyon* and wheat by pathogen *F. culmorum*.** The main experimental factor is previous inoculation with the endophytic fungus *M. bolleyi*. CI– 95% confidence intervals. α = 0.05, statistically significant difference marked with asterisk (*).

| Species | Endophyte | n | Mean leaf damage (%) | Standard error | CI −95% | CI +95% |
|---------|-----------|---|----------------------|----------------|---------|---------|
| Bd | Mb1 (yes) | 30 | 14.33 | 4.2259 | 5.6904 | 22.9762 |
| | Mb0 (no) | 30 | 51.83* | 5.6603 | 40.2566 | 63.4100 |
| Wheat | Mb1 (yes) | 30 | 16.16 | 4.8345 | 6.2789 | 26.0543 |
| | Mb0 (no) | 30 | 46.50* | 5.9458 | 34.3393 | 58.6606 |

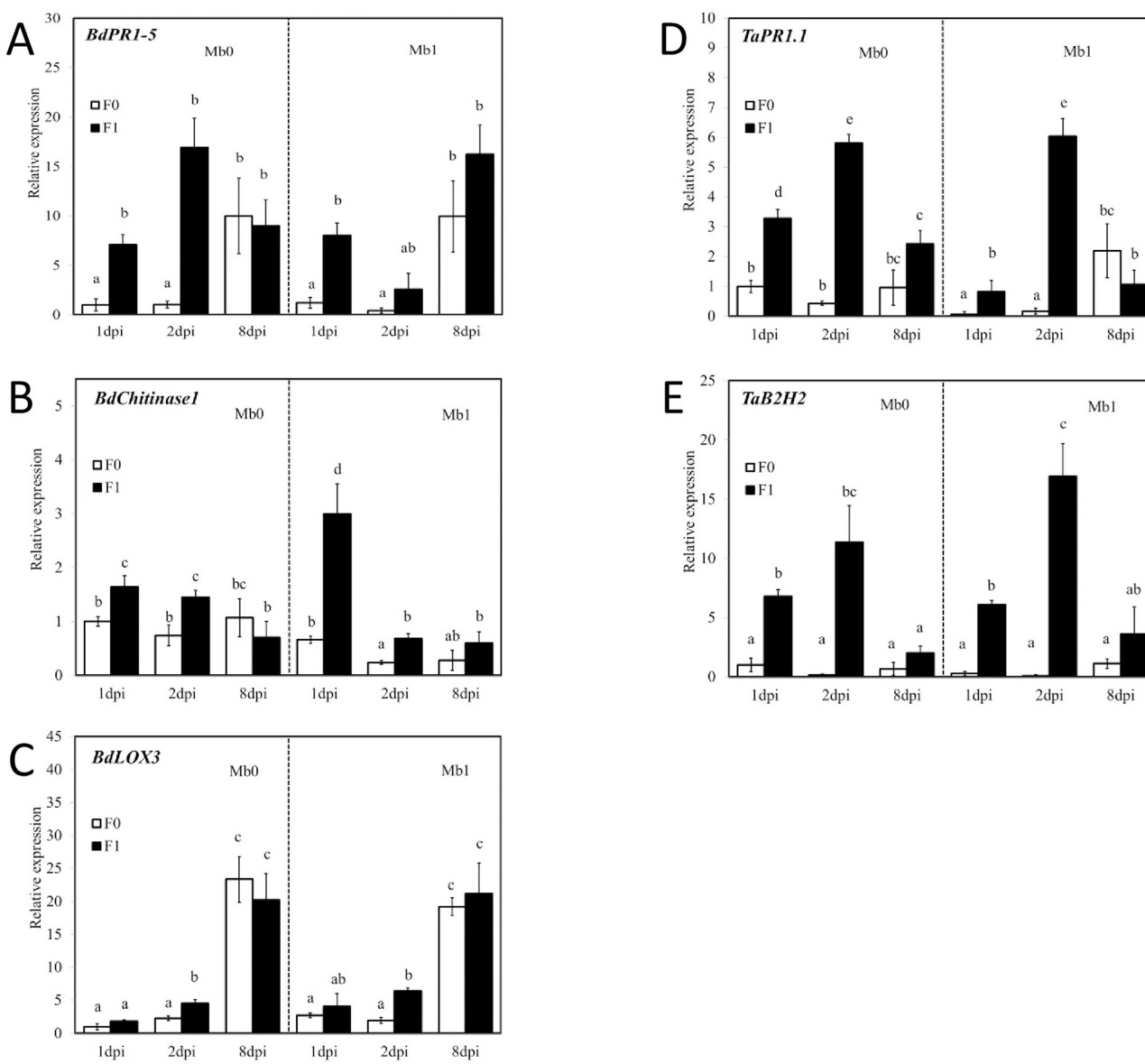

**Fig 2. Expression profiles of genes *BdPR1-5* (A), *BdChitinase1* (B) and *BdLOX3* (C) in *Brachypodium distachyon* (Bd) and *TaPR1.1* (D) and *TaB2H2* (E) in wheat inoculated with *Microdochium bolleyi* (Mb1) or not so inoculated (Mb0) and with leaves infected with *Fusarium culmorum* (F1) or not so infected (F0) at three time periods 1, 2, and 8 days post inoculation (dpi).** Expression levels are relative to *M. bolleyi* non-inoculated and Fc non-infected leaves of plants at 1 dpi and were normalized with the reference genes *BdSamDC* and *TaGADPH*. Expression levels shown are mean values and standard deviations (whiskers) for three replications. Statistically significant differences are indicated by letters above columns (post hoc Tukey's test, *P* < 0.05).

FD = 4.12. Other increases were FD = 14.26 for *TaPR1.1* and FD = 39.66 for *TaB2H2*. In these genes, the changes were clear at 1 dpi and 2 dpi, but there were no conclusive differences at the last sampling date of 8 dpi. In the case of *BdLOX3*, the difference due to pathogen infection was also apparent at 1 dpi and 2 dpi (FD = 8.8), and no difference was noted later. Nevertheless, a statistically significant difference was observed only at 2 dpi (Fig 2).

Inoculation of plants with the endophytic fungus Mb did not affect the expression of tested marker genes (Table 3). There were no differences in the expression of the genes studied when

**Table 3. Expression-level differences for chosen genes in leaves of *Brachypodium distachyon* (Bd) and wheat plants.** Gene expression was detected as fold difference (FD) between inoculated (by pathogen or endophyte) and non-inoculated plants (mock).

| Gene | Pathogen | | | Endophyte | | |
|------|----------|---|---|-----------|---|---|
| | Mean FD | *F*-stat | *P*-value | Mean FD | *F*-stat | *P*-value |
| *BdPR1-5* | 30.944 | 16.664 | 0.00129 | −13.795 | 0.823 | 0.38090 |
| *BdChitinase1* | 4.120 | 9.365 | 0.00847 | −0.260 | 0.018 | 0.89457 |
| *BdLOX3* | 8.807 | 10.507 | 0.00590 | 5.493 | 0.085 | 0.77299 |
| *TaPR1.1* | 14.267 | 18.859 | 0.00067 | −3.431 | 0.490 | 0.49554 |
| *TaB2H2* | 39.661 | 32.744 | 0.00005 | 4.041 | 0.094 | 0.76409 |
| Mean | 19.559 | | | −1.590 | | |
| Total | 97.799 | | | −7.952 | | |

The table shows means calculated from 1 dpi and 2 dpi data. Data from 8 dpi was excluded because there were no statistically significant differences among treatments. Non-significant differences were found in FD for the genes *BdPAL*, *BdEIN3*, *BdAOS*, *TaPAL*, *TaLOX*, *TaEIN2*, *TaAOS* and so these are not displayed in the table. A statistically significant difference is expressed by a *P*-value lower than 0.05.

Mb and Fc treatments were compared with treatment with a single pathogen infection (Table 4), except in two cases. Mb affected the expression of *BdChitinase1* and *TaPR1.1* genes at 1 dpi (Fig 2B and 2D). While *BdChitinase1* was upregulated in Mb1 treatment (with endophyte) at 1dpi, *TaPR1.1* expression was downregulated at the same term. The expression of the other genes tested was not affected by Mb inoculation.

## Discussion

Root endophytes can confer resistance against plant pathogens by direct antagonism through mycoparasitism, antibiosis, and/or competition for nutrients and niches, or indirectly by triggering induced resistance in the host [50, 51]. Indirect antagonism takes into account an effect of the endophyte on the pathogen mediated via the host plant by triggering induced systemic resistance (ISR). Endophytes trigger ISR by two phytohormones, jasmonic acid (JA) and ethylene (ET), thereby resulting in a faster and stronger immune response following pathogen attack. Some endophytes are able to trigger a systemic acquired resistance response (SAR) that is salicylic acid (SA) dependent and also results in primed host defences [50, 52, 53].

**Table 4. Expression-level differences for chosen genes in leaves of *Brachypodium distachyon* (Bd) and wheat plants.** Gene expression was detected as fold difference (FD) between plants inoculated only by pathogen and plants inoculated both by endophyte and pathogen to evaluate the effect potentially caused by endophyte on subsequent pathogen infection.

| Gene | Mean FD | *F*-stat | *P*-value |
|------|---------|----------|-----------|
| *BdPR1-5* | −1.307 | 1.084 | 0.33803 |
| *BdChitinase1* | 0.398 | 0.176 | 0.68937 |
| *BdLOX3* | 2.247 | 3.692 | 0.10305 |
| *TaPR1.1* | −0.678 | 0.446 | 0.52917 |
| *TaB2H2* | 0.717 | 0.415 | 0.54309 |
| Mean | 0.275 | | |
| Total | 1.376 | | |

Table shows means calculated from 1 dpi and 2 dpi. A statistically significant difference is expressed by a *P*-value lower than 0.05.

Most previous research with monocotyledonous plants has been focused on agronomically important grasses, such as tall fescue (*Festuca arundinacea*), perennial ryegrass (*Lolium perenne*), and meadow fescue (*Festuca pratensis*), and their interactions with herbivores and pathogens [54, 55]. Endophytic fungi of the genus *Epichloë* and their asexual *Neotyphodium* forms are thought to interact mutualistically with their host grasses, providing protection for the host against herbivores and pathogens mediated by fungal alkaloids [56]. There are many other examples of endophytes that decrease disease susceptibility of their host upon pathogen infection [50, 57, 58], thus making endophytes useful agents for disease control [50]. Endophytic fungus *Harpophora oryzae* in rice roots protected rice from *Magnoporte oryzea* root invasion by the accumulation of $H_2O_2$ and elevated antioxidative capacity. *H. oryzae* also induced systemic resistance against rice blast by upregulation expression of transcription factor *OsWRKY45* [59]. Inoculation of *Theobroma cacao* leaves with endophytic fungus *Colletotrichum tropicale* leads to higher resistance to damage by the pathogen *Phytophthora palmivora* [60], lower incidence of black pod disease caused by *Phytophthora* spp. [61] moreover, it induces changes in the expression of host genes with defense responses functions. Inoculations of *T. cacao* by endophyte *C. tropicale* produced changes in the expression of genes associated with the synthesis, modification, and degradation of the cell wall; peroxidases and components of the jasmonic acid defense pathway; pathogenesis-related proteins; redox state proteins; genes coding beta glucanases, heat shock proteins, transcription factors, proteins involved in secondary metabolism and proteolysis; and other genes relevant to plant-microbe interactions such as NPR3, nodulin, and endochitinases [62].

In our work, plants colonized by Mb showed significantly lower levels of Fc infection. Fc infection induced expression of *BdPR1-5*, *BdChitinase1*, *BdLOX3*, *TaB2H2*, and *TaPR1.1*, but this expression was not affected by previous inoculation with the Mb endophyte. In cases of indirect endophyte-mediated resistance, a typical SAR/ISR response is based upon initial exposure to an endophyte that primes plant defence and subsequent infection with the pathogen [50]. The phenotypic resistance observed in this study, together with the gene expression findings in Bd and wheat, suggests that other genes are involved in the putative endophyte-mediated resistance. However, further research is needed to demonstrate which genes are responsible for this endophyte-mediated resistance. Since Mb is a root endophyte, mechanisms such as direct antimicrobial activity or competition for niches or resources cannot be assumed to be involved in the increased host resistance observed on leaves. Similarly, a study by Constantin *et al.* on tomato demonstrated that fungal endophyte *Fusarium oxysporum* strain Fo47 can confer endophyte-mediated resistance independently of SA, ET, or JA [50]. Their findings imply that endophyte-mediated resistance is mechanistically distinct from ISR and SAR.

## Expression of marker genes

Striving to elucidate the possible factors involved in endophyte-mediated resistance in Bd, the activity of genes known to be involved in the activation of defence-related pathways in response to pathogen attack were measured. In model plants, genes considered to be involved in phytohormone biosynthesis or signalling are also used as markers of plant responses to pathogen attack [63]. Many defence genes are involved in SA, JA, and ET signalling pathways and are known to be associated with plant defence generally and with such model plants as *Arabidopsis* [64] and Bd [65]. JA- and ET-mediated signalling pathways are mainly linked to plant responses to necrotrophic pathogens and herbivores, while the SA-dependent pathway is mainly involved in responses to biotrophic pathogens, and these pathways can act antagonistically [66]. Hormone-responsive genes are thus used to evaluate disease resistance responses

during pathogen infection. Defence-related phytohormone marker genes expressed at time points suitable for defence-response monitoring were studied for Bd by Kouzai et al. [65] and by Sandoya et al. (1). Some of the recommended marker genes from the two studies just cited were used in the current study to evaluate response of Bd to endophyte fungus Mb and also pathogenic Fc infection. To confirm the hypothesis, the influence of the endophyte Mb on wheat was investigated in parallel.

Pathogenesis-related (PR) genes generally encode small antimicrobial proteins, their expression levels increase quickly when stimulated by biotic or abiotic stress, and therefore they serve as markers for plant immune signalling. PR proteins' biochemical activity and mode of action have nevertheless remained elusive [67]. PR-1 proteins constitute a large family of 23 proteins in wheat that are upregulated early in the defence response [68]. Findings from a recent study provided genetic and biochemical evidence indicating that *PR1* binds to sterols and inhibits pathogen growth [69]. Increased expression of the *PR1*, *PR2*, and *PR5* genes represents activation of the SA signalling pathway [70], but, in rice, for example, the expression level of *OsPR1* is increased by both SA and JA [71]. Although expression of *PR1.1* and *PR1.2* genes of wheat has been induced upon infection with either compatible or incompatible isolates of the fungal pathogen *Blumeria graminis*, these genes did not respond to such activators of systemic acquired resistance (SAR) as SA, benzothiadiazole, or isonicotinic acid [72]. Similarly in the present study, *PR1* was induced by Fc but not by endophyte-mediated resistance provided by pre-inoculation of Bd and wheat with Mb. In a study by Zhang *et al.*, two PR protein-encoding genes (*PR1–17* and *PR10*) showed progressive increase in expression over time that peaked after the appearance of symptoms in *G. graminis* var. *tritici*-infected roots of wheat [73]. In our study, the gene *PR1.1* in wheat was highly expressed at 1 dpi and 2 dpi after Fc infection. In retrospect, in our study, it would have been better to measure the expression of genes earlier, within a matter of hours (1, 2, 3 h, etc.), after infection. After 24 h, the expression of some genes may already have passed its peak. There were no longer any statistically significant differences in the level of *PR1* gene expression between the Fc infected and non-infected treatments in either Bd or wheat hosts at 8 dpi.

Jasmonic acid is an oxylipin hormone derived from linolenic acid that is crucial for plants to regulate growth and development, as well as to respond to biotic and abiotic stresses [74]. Allene oxide synthase (*AOS*) and lipoxygenase (*LOX*) are required for JA biosynthesis [75], and they are used as JA markers in various plant species [65]. In a study by Zhang *et al.* [73], *G. graminis* var. *tritici*-infected roots showed changes in genes expression profile after pathogen infection. *LOX2* had a unique expression pattern in that it progressively declined after infection. Similarly, in our study *LOX3* expression in Bd after Fc infection was significantly elevated at 1 and 2 dpi. *AOS* usually reacts to wounding of plant organs, as described, for example, in *Arabidopsis* or flax [76, 77] or to feeding damage by herbivores in rice [78]. Plants also react through a change in *AOS* expression to the effect of attack by pathogenic fungi. Various wheat cultivars infected with the fungal pathogen *Zymoseptoria tritici* have shown significant changes in *AOS* and *LOX* expression levels in comparison with non-infected controls [79]. In our study, neither inoculation with the endophyte nor with Fc demonstrated any effect on *AOS* expression either in Bd or in wheat. In a study by Gottwald *et al.*, defence-related genes of wheat encoding jasmonate-regulated proteins were upregulated in response to FHB [80]. The transcription levels of the *LOX3* gene in the JA pathway of Bd studied by Sandoya *et al.* (1) were upregulated in the resistant genotype at 24 h post infection, with a decrease observed at 48 h following infection with fungal pathogen *Sclerotinia homoeocarpa* isolate.

Phenylalanine ammonia lyase (*PAL*) is a key enzyme in the phenylpropanoid pathway of higher plants, and transcriptional upregulation of *PAL* after pathogen infection has been reported in *Arabidopsis*, wheat, rice, and *Brachypodium* [5, 81]. From a plant defence

viewpoint, *PAL* is responsible for synthesizing such compounds as flavonoids and chlorogenic acid [82]. A study by Cass *et al.* [83] showed that *PAL* in Bd plays an important role in lignin biosynthesis, with more than 40% reduction in cell wall lignin content associated with *PAL* knockdown. Reduced-lignin plants exhibited significantly increased susceptibility to the hemi-biotrophic fungal pathogens Fc and *Magnaporthe oryzae* [83]. There were no changes in expression of the *PAL* gene after Fc and Mb inoculation in either Bd or wheat in the current study.

Ethylene regulates many diverse metabolic and developmental processes in plants, ranging from seed germination to senescence, and it is considered to play an important role as a signal molecule during abiotic and biotic stress [84, 85]. There was measured gene expression of *EIN2* (ethylene-insensitive 2) in wheat and *EIN3* (ethylene-insensitive 3) in Bd, which encode *EIN*, a small nuclear-localized protein that is considered a transcription factor acting as a positive regulator in the ethylene response pathway. It is required for ethylene responsiveness in adult plant tissues and binds a primary ethylene response element present in the *ethylene-response-factor 1* (*ERF1*) promoter with the consequence of activating transcription of this gene [86, 87]. *EIN3* is also involved in a regulatory cascade for the modulation of PR genes expression [88]. In the interaction of *Arabidopsis thaliana* and bacterial pathogen *Erwinia amylovora*, the elicitor HrpNEa activates ethylene-mediated expression of the *Arabidopsis* transcription factor *MYB44*, which in turn enhances the expression of *ethylene-insensitive 2* (*EIN2*) [89]. No changes in *EIN* gene expression after Fc and Mb inoculation were observed in either Bd or wheat in the current study. JA and ET are two major plant hormones that synergistically regulate plant development and tolerance to necrotrophic fungi. *EIN3* and *EIL1* are positive regulators of a subset of JA responses, including PR gene expression, plant resistance to necrotrophic fungi, and root development [90]. The endophyte must use enzymes to overcome obstacles while growing through the plant tissues and the plant duly responds to these signals by its defence mechanisms, much as it would react to a pathogen and its virulence factors. How this delicate balance is regulated remains a question, however.

## Endophyte diagnosis by PCR

Endophyte presence is usually determined by methods other than PCR, especially microscopic [43] and by culture [91, 92]. In recent years, molecular methods have also been introduced [93]. Ernst *et al.* [94] used nested PCR to distinguish two endophytic species of Mb and *Microdochium phragmitis* in *Phragmites australis*. In the present study, primers for end-point PCR to determine the Mb endophytic fungus in plant tissues were developed. Although endophytes had previously been determined by PCR in grass tissues, such as in cases of *Epichloë* [95], *Neotyphodium* [96], and *Acremonium* [97], no method has been available heretofore for standard end-point PCR determination of Mb. The method was tested in wheat and Bd. In both host species, the presence of Mb could be unequivocally demonstrated in the inoculated plants by the presence of a band on the electrophoretic gel. In plants without previous Mb inoculation, the results of PCR test using the MbPOLII primer were negative. Therefore, the molecular method is a suitable complement to, or might completely replace, other tests for detecting Mb in plant tissues. The results from studies of endophytic fungi to date are in many aspects still fragmentary, and nobody has yet managed, for instance, to cover the entire spectrum within different parts of a plant simultaneously, let alone across several seasons. This is mainly due to methodological limitations of the endophyte studies. This PCR-based method of endophyte determination facilitates such research in many respects.

## Conclusion

Bd is a suitable model for studying interactions with the endophytic fungus Mb, and it is possible to study the influence of endophyte-mediated resistance at the phenotypic level as well as in terms of gene expression. Bd and Mb showed positive interaction. In plants colonized by Mb, less severe symptoms of damage created by the fungus Fc were detected. *Chitinase*, *PR1*, and *LOX* in Bd as well as *B2H2* and *PR1.1* in wheat, reacted to Fc by their upregulation. In terms of endophyte-mediated resistance, these genes were in most cases independent of Mb's presence. There were developed primers for the end-point PCR to diagnose the endophyte Mb in plant tissues and tested it in Bd and wheat. The method is suitable for confirming presence of the endophyte in plants.

## Supporting information

**S1 Fig. Schema for evaluating percentage necrotization on plant leaves wounded by Pasteur pipette and infected by *F. culmorum* or wounded and mock inoculated.**
(TIF)

**S2 Fig. Wheat roots with rhizodermal cells filled by chlamydospores of *Microdochium bolleyi*.** Cross-section by wheat root with cells filled by chlamydospores (arrowed), bar 100μm (A). Wheat rhizodermis with root cells filled by chlamydospores, bar 100μm (B).
(PDF)

**S3 Fig. Sensitivity test of PCR with primers MbPOLIIF/R on five amounts of Mb DNA (ng).** Three different Mb isolates were tested: A) UPOC-FUN-253, B) UPOC-FUN-254, and C) UPOC-FUN-255.
(TIF)

**S4 Fig. Macroconidia of *F. culmorum* germinating and dark-coloured necrotization of wheat tissue.** Asterisks indicate light-coloured areas originating from wounding by Pasteur pipette prior to infection. Arrows indicate some germinated macroconidia. (bar = 50 μm).
(TIF)

**S5 Fig. Leaves of *Brachypodium distachyon* after bleaching for 3 days in 2.5% KOH.** A) Non-inoculated plants with endophyte and without infection by Fc (Mb0F0), B) Non-inoculated plants with endophyte and with infection by Fc (Mb0F1). C) Leaves of plants inoculated with endophyte and infected with Fc (Mb1F1), and D) leaves of plants inoculated with endophyte and without Fc infection (Mb1F0).
(TIF)

**S1 Table. Primer pairs used in the study.** Names, sequences of forward and reverse primers, publication sources of primer pairs, and gene functions are listed.
(DOCX)

**S2 Table. Microscopic and molecular detection of Mb in Bd and wheat roots.** The microscopic method was based on the presence of chlamydospores in roots after staining with aniline blue. PCR methods were performed using MbITSRTF/R primers.
(DOCX)

**S3 Table. ANOVA statistical evaluation for damage to leaves of *Brachypodium distachyon* by pathogen *Fusarium culmorum* when the main experimental factor is previous inoculation with the endophytic fungus *Microdochium bolleyi*.**
(DOCX)

**S4 Table. ANOVA statistical evaluation for damage to leaves of wheat by pathogen *Fusarium culmorum* when the main experimental factor is previous inoculation with the endophytic fungus *Microdochium bolleyi*.**
(DOCX)

## Acknowledgments

We thank the Collection of Phytopathogenic Microorganisms UPOC for providing fungal isolates.

## Author Contributions

**Conceptualization:** Pavel Matušinsky, Dominik Bleša.

**Data curation:** Pavel Matušinsky.

**Formal analysis:** Pavel Matušinsky, Dominik Bleša.

**Funding acquisition:** Pavel Matušinsky.

**Investigation:** Pavel Matušinsky, Dominik Bleša.

**Methodology:** Pavel Matušinsky, Dominik Bleša.

**Project administration:** Pavel Matušinsky.

**Resources:** Pavel Matušinsky.

**Software:** Pavel Matušinsky, Dominik Bleša.

**Supervision:** Pavel Matušinsky.

**Validation:** Pavel Matušinsky, Božena Sedláková.

**Visualization:** Pavel Matušinsky, Dominik Bleša.

**Writing – original draft:** Pavel Matušinsky, Dominik Bleša.

**Writing – review & editing:** Pavel Matušinsky.

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
