## [Decision Letter · Decision Letter 0]

14 Nov 2021

PONE-D-21-30855Compatible interaction of Brachypodium distachyon and endophytic fungus Microdochium bolleyiPLOS ONE

Dear Dr. Matušinsky,

Thank you for submitting your manuscript to PLOS ONE. After careful consideration, we feel that it has merit but does not fully meet PLOS ONE’s publication criteria as it currently stands. Therefore, we invite you to submit a revised version of the manuscript that addresses the points raised during the review process.

The paper has been revised by two experts who both suggested major revision in order to improve the ms and render it acceptable for publication in PlosOne. Authora are invited to carefully follow reviewers' suggestion and to submit a revised version of their work.

We look forward to receiving your revised manuscript.

Kind regards,

Sabrina Sarrocco

Academic Editor

PLOS ONE

Journal Requirements:

- https://www.frontiersin.org/articles/10.3389/fpls.2019.00979/full

The text that needs to be addressed involves the Discussion section.

In your revision ensure you cite all your sources (including your own works), and quote or rephrase any duplicated text outside the methods section. Further consideration is dependent on these concerns being addressed.

Reviewers' comments:

Reviewer's Responses to Questions

**Comments to the Author**

1. Is the manuscript technically sound, and do the data support the conclusions?

Reviewer #1: Partly

Reviewer #2: Partly

2. Has the statistical analysis been performed appropriately and rigorously? 

Reviewer #1: Yes

Reviewer #2: Yes

3. Have the authors made all data underlying the findings in their manuscript fully available?

Reviewer #1: Yes

Reviewer #2: No

4. Is the manuscript presented in an intelligible fashion and written in standard English?

Reviewer #1: Yes

Reviewer #2: No

5. Review Comments to the Author

Reviewer #1: Dear corresponding author,

The manuscript number PONE-D-21-30855 titled “Compatible interaction of Brachypodium distachyon and endophytic fungus Microdochium bolley” investigate the effect of this endophytic fungus on the intensity of the attack by pathogen Fusarium culmorum in B.distachyon and wheat and tested changes in expression of genes (in B. distachyon: BdChitinase1, BdPR1, BdLOX3, BdPAL, BdEIN3, and BdAOS; and in wheat: TaB2H2(chitinase), TaPR1, TaLOX, TaPAL, TaEIN2, and TaAOS) involved in defense against pathogens. The nature of the subject studied and the results obtained are worthy to be taken into account for publication in PLOS ONE. However, some aspects need to be clarified and a revision is necessary to increase the article quality. For this reason, I provided a revision list. I hope that my considerations could be useful to improve your study and its clarity to the reader.

Revision list:

1) Lines 1-2: About the title, I was wondering if is possible to speak of “compatible interaction” between a host and an endophytic fungus such as Microdochium bolleyi. In addition why you did not mention also wheat?

2) ABSTRACT: please avoid the excessive use of “we” in the abstract section as well as in other parts of the manuscript;

3) Line 23: Please change infestation with infection, infestation is not correct for pathogenic fungi;

4) Line 26-29: Please reword this sentence because is not clear to the reader;

5) Line 52: Please describe better the distribution of F. culmorum in the different world regions;

6) Lines 65-67: Please provide more details of the activity of Microdochium bolleyi especially against F. culmorum;

7) Lines 68-73: The aim of the work is not clear. Please reformulate it;

8) Line 82: Please add more detail about Microdochium bolleyi isolates, in particular, host of origin and country of origin;

9) TABLE 1: What are MbPOLII? Is this the first time that this abbreviation or name appears, please explain;

10) Line 108: Please provide more details of inoculum preparation, because is not enough what has been provided;

11) Line 109: Why you used the word “variants”?

12) Line 137: Fungal mycelia of what species? Please specify;

13) Line 139: Only roots or also leaves?

14) Lines 147-148: I suggest inserting the results of the preliminary screen in the main text and not as supporting information because is very important for primer pairs design;

15) Line 159: please change the expression “vis-à-vis”;

16) Lines 174-175: please explain the choice of plant defense genes, in other words, why did you choose these genes?

17) Lines 205-206: Again please insert in the main text this information (see comment number 14);

18) Lines 211-212: Where these results are reported?

19) Lines 231-278: All the paragraph “expression levels of genes involved in plant-pathogen interaction in Bd and wheat” is not clear. Please try to re-organize sentences and table clearer;

20) Lines 294: Please mention briefly these examples;

21) Lines 300-301: Please provide an accurate hypothesis relative to other mechanisms involved in the phenotypic resistance observed in this study;

22) Lines 391-392: Please be sure about this sentence;

23) Lines 412-413: The last sentence of the paper implies that all the sections relative to the Microdochium bolleyi identification method should be more accurate. So please also based on previous comments (for example 14 and 17) edit with more attention all aspects of Microdochium bolleyi identification because you are going to propose a method of identification that should be very accurate and precise.

Best regards.

Reviewer #2: The manuscript from Matušinsky et al. studies the impact of Microdochium bolleyi (Mb) on both Brachypodium and wheat leaves infection by Fusarium culmorum (Fc). First, the authors aim establish the endophytic status of Mb in roots both through microscopic observations and by molecular methods. Then, using a set of defense marker genes, they try to decipher whether Mb induces the expression of defense genes.

The study is overall well conducted with appropriate controls and replicates. The manuscript is clear and well-structured.

Nevertheless, there are two weak points in the manuscript that prevent acceptance of the manuscript and which should be improved. The first point concerns whether Mb is a true endophyte in Brachypodium (or wheat) roots. The second point is the relationship between Mb potential endophytism and the induction of defense genes.

Endophytism of Mb in roots is, to my opinion, insufficiently demonstrated. Indeed, microscopic observations are not fully convincing. Pictures (Fig1) are not sufficient to fully demonstrate the colonization of Brachypodium roots by Mb or the presence of chalmydospores within host root cells. The authors should provide cross-sections to fully demonstrate that Mb is truly inside the roots. In the same part, the authors claim they manage to detect Mb by PCR. If PCR results using pure Mb DNA and primers MbPOLIIF/R are indeed statisfying, no data are presented on roots. Moreover, in the Material and Methods section, the authors do not mention any surface sterilization, which is a very important step in endophytism studies. This part should be strengthened to fully demonstrate that Mb is indeed an endophytic fungus.

The second aspect that needs to be improved concerns the study of defense genes expression. The first point is that the authors seem not to use the same biological material to test endophytism (in roots) and defense gene expression (in leaves). Indeed, they mention that Fc inoculations are performed “In the phase of the second offshoot in Bd and third leaf of wheat » which, albeit not being very precise, means rougly three-week old plants, whereas endophytism was tested 90 days after sowing that is much later. In this context, it is very difficult to compare expression analysis with the putative endophytic status of Mb as no result establishes that Mb is truly endophytic at the stage of inoculation by Fc.

Additional comments

Lines 240-256: the whole paragraph is unclear. The authors should first describe the induction of gene expression by Fc infection then describe how Mb presence may modify gene expression.

Lines 242-243: BdChitinase1 and TaPR1.1 do show differential expression in the presence of Mb at 1 dpi. Why do the authors claim no statistical difference in gene expression

Minor revisions

Line 85: the 6 isolates were mixed together, detail in which proportion

Lines 234-235: gene names should be revised as used in the cited references (e.g. BdPR1 is BdPR1.1 in Kouzai et al. 2016)

6. PLOS authors have the option to publish the peer review history of their article (what does this mean?). If published, this will include your full peer review and any attached files.

Reviewer #1: No

Reviewer #2: No

---

## [Author Response · Author response to Decision Letter 0]

22 Dec 2021

Reviewer #1: Dear corresponding author. The manuscript number PONE-D-21-30855 titled “Compatible interaction of Brachypodium distachyon and endophytic fungus Microdochium bolley” investigate the effect of this endophytic fungus on the intensity of the attack by pathogen Fusarium culmorum in B.distachyon and wheat and tested changes in expression of genes (in B. distachyon: BdChitinase1, BdPR1, BdLOX3, BdPAL, BdEIN3, and BdAOS; and in wheat: TaB2H2(chitinase), TaPR1, TaLOX, TaPAL, TaEIN2, and TaAOS) involved in defense against pathogens. The nature of the subject studied and the results obtained are worthy to be taken into account for publication in PLOS ONE. However, some aspects need to be clarified and a revision is necessary to increase the article quality. For this reason, I provided a revision list. I hope that my considerations could be useful to improve your study and its clarity to the reader.

Dear Reviewer 

We thank the reviewer for all comments. We have accepted all suggestions for changes and tried to implement them as best as we can. Below is a detailed description of our responses to the suggestions. We sincerely hope that the article is now clearer for readers.

Revision list:

1) Lines 1-2: About the title, I was wondering if is possible to speak of “compatible interaction” between a host and an endophytic fungus such as Microdochium bolleyi. In addition why you did not mention also wheat?

We were looking for the best title, short and clear. It is also possible to use the terms "Host Affinity to Endophyte" or "Successful Colonization" or "Positive Association" or other synonyms. Perhaps we could write only "Interaction of Brachypodium distachyon and the endophytic fungus Microdochium bolleyi," but in this case, the critical fact that the interaction is successful would be missing from the title. Therefore, we have used the term "compatible interaction", which is used in the case of positive interactions between host and pathogen, and hope this is concise and clear. Nevertheless, we are ready to change the title if necessary.

The main character of the story is Bd. Therefore, the title favours Bd, and likewise, most of the text focuses on Bd. However, we are aware that Bd is "only" a model for wheat and cereals in general, so we have paralleled wheat in the experiments with Bd. What is already known in many previous examples, and what pleased us during this study, is that the observed phenomena, such as endophyte and pathogen responses, were very similar in both host species, and therefore Bd is a suitable model.

2) ABSTRACT: please avoid the excessive use of “we” in the abstract section as well as in other parts of the manuscript;

We thank the reviewer for this comment, and we corrected it in the whole manuscript. 

3) Line 23: Please change infestation with infection, infestation is not correct for pathogenic fungi;

We thank the reviewer for this comment, and we corrected it.

4) Line 26-29: Please reword this sentence because is not clear to the reader;

We corrected this sentence to make it more transparent.

5) Line 52: Please describe better the distribution of F. culmorum in the different world regions;

We thank the reviewer for this comment, and we added information to the new version of manuscript.

6) Lines 65-67: Please provide more details of the activity of Microdochium bolleyi especially against F. culmorum;

We thank the reviewer for this comment, and we added information to the new version of manuscript.

7) Lines 68-73: The aim of the work is not clear. Please reformulate it;

We thank the reviewer for this comment; we rewrote the whole paragraph.

8) Line 82: Please add more detail about Microdochium bolleyi isolates, in particular, host of origin and country of origin;

We added the required information.

9) TABLE 1: What are MbPOLII? Is this the first time that this abbreviation or name appears, please explain;

We added the required information.

10) Line 108: Please provide more details of inoculum preparation, because is not enough what has been provided;

We added the required information.

11) Line 109: Why you used the word “variants”?

We thank the reviewer for this comment, and we corrected it. The word “variants” was replaced by “treatments” in the whole manuscript. 

12) Line 137: Fungal mycelia of what species? Please specify;

We added the required information.

13) Line 139: Only roots or also leaves?

To determine M. bolleyi DNA in plants, DNA was isolated from roots only, as M. bolleyi is exclusively a root endophyte. Leaves were used for RNA isolation; please see the next chapter.

14) Lines 147-148: I suggest inserting the results of the preliminary screen in the main text and not as supporting information because is very important for primer pairs design;

We have added information about other primers from the pre-screening.

15) Line 159: please change the expression “vis-à-vis”;

The word “vis-à-vis” was replaced by “towards”.

16) Lines 174-175: please explain the choice of plant defense genes, in other words, why did you choose these genes?

The genes were selected primarily regarding the possibility of testing the plant defence marker genes in Bd. We selected these genes because they have been validated and well characterized in previous publications (Kouzai et al 2016; Sandoya et al 2014; Hong et al 2008). Based on this selection, we then analogously selected genes for wheat. There are multiple options in wheat, so the availability of validated and well-established genes of Bd was vital for selection.

17) Lines 205-206: Again please insert in the main text this information (see comment number 14);

We have added information about other primers from the pre-screening.

18) Lines 211-212: Where these results are reported?

We added these results to the Supporting information in new version of manuscript; please see Table S4. We decided to put this table to the supporting information and not to the main text because of the table's simplicity. All samples inoculated were positive, and all samples non-inoculated were negative. In our view, the description of results should be in the main text sufficient and the supporting table in supplementary.

19) Lines 231-278: All the paragraph “expression levels of genes involved in plant-pathogen interaction in Bd and wheat” is not clear. Please try to re-organize sentences and table clearer;

We have reorganized the entire paragraph. In the new version of the manuscript, we described first the pathogen's effect, then the endophyte's effect, and finally, the effect of the endophyte on plants infected by the pathogen. In the new version of the article, we, therefore, split the original table. We are now presenting the assessment of the effect of endophyte on gene expression in hosts co-infected with the pathogen in a separate table, and we believe that the text is now much clearer.

20) Lines 294: Please mention briefly these examples;

We have added more examples to the new version of the manuscript. We added the example of Rice – Harpophora - Magnoporte, where both the increase in resistance to the pathogen due to endophyte and the effect on the increase of the gene expression of the transcription factor is documented. We have also added a Cacao – Colletotrichum - Phytophora example where it is also described how endophyte reduces symptoms and which genes are affected. Another example, which was also in the original version of the manuscript, is the tomato - Fo47 - F. oxysporum system where again the reduction of symptoms is described and the effect on the expression of genes involved in plant defence is discussed.

21) Lines 300-301: Please provide an accurate hypothesis relative to other mechanisms involved in the phenotypic resistance observed in this study;

Thanks to the reviewer for this comment. Mb is a root endophyte, mechanisms such as antimicrobial activity or niche or nutrient competition are unlikely to be responsible for endophyte-mediated resistance. We have included this hypothesis in the manuscript. We also speculate that endophyte affects other genes, but the information is scarce, and further research is needed. We hope that this paper's new Bd-Mb system and new diagnostic tool we are bringing to the scientific community will make this challenge more available.

22) Lines 391-392: Please be sure about this sentence;

It is true that this sentence has no scientific value for the article and has therefore been removed.

23) Lines 412-413: The last sentence of the paper implies that all the sections relative to the Microdochium bolleyi identification method should be more accurate. So please also based on previous comments (for example 14 and 17) edit with more attention all aspects of Microdochium bolleyi identification because you are going to propose a method of identification that should be very accurate and precise.

Best regards.

We thank the reviewer for this comment and we accept this opinion. We have added information on other tested combinations of primers based on ITS, LSU (large subunit of ribosomal gene), Tub2 (β-tubulin gene), and RPB2 (RNA polymerase II second-largest subunit) to the new version of manuscript. We have also added a table showing the results of the primer assays in Mb inoculated and non-inoculated plants (Table S4). The quality of the final MbPOLIIF/R primers is supported by a sensitivity test on M. bolleyi DNA, a specificity test on a large set of species potentially occurring fungi in the samples, and a test of the applicability of Mb endophyte diagnostics in host tissues on two species (Bd and wheat).

Reviewer #2: The manuscript from Matušinsky et al. studies the impact of Microdochium bolleyi (Mb) on both Brachypodium and wheat leaves infection by Fusarium culmorum (Fc). First, the authors aim establish the endophytic status of Mb in roots both through microscopic observations and by molecular methods. Then, using a set of defense marker genes, they try to decipher whether Mb induces the expression of defense genes. The study is overall well conducted with appropriate controls and replicates. The manuscript is clear and well-structured. Nevertheless, there are two weak points in the manuscript that prevent acceptance of the manuscript and which should be improved. The first point concerns whether Mb is a true endophyte in Brachypodium (or wheat) roots. The second point is the relationship between Mb potential endophytism and the induction of defense genes.

Dear Reviewer 

We thank the reviewer for all comments. We have accepted all suggestions for changes and tried to implement them as best as we can. We considered two main points concerning the nature of true Mb endophytism and the potential of endophyte-inducing defense genes. Our responses to these questions are detailed below. We sincerely hope that the article is now clearer for readers.

Endophytism of Mb in roots is, to my opinion, insufficiently demonstrated. Indeed, microscopic observations are not fully convincing. Pictures (Fig1) are not sufficient to fully demonstrate the colonization of Brachypodium roots by Mb or the presence of chalmydospores within host root cells. The authors should provide cross-sections to fully demonstrate that Mb is truly inside the roots. In the same part, the authors claim they manage to detect Mb by PCR. If PCR results using pure Mb DNA and primers MbPOLIIF/R are indeed statisfying, no data are presented on roots. Moreover, in the Material and Methods section, the authors do not mention any surface sterilization, which is a very important step in endophytism studies. This part should be strengthened to fully demonstrate that Mb is indeed an endophytic fungus.

At the request of reviewer, we have made a cross-section of roots and added figures to the appendix (Fig. S2). These images show colonization of epidermal cells of wheat roots by Mb chlamydospores. The figure of cross-section by wheat roots was added to the manuscript, but the figure of cross-section of Bd we showed here only for reviewer information (see below in this document and in attachment for reviewer). We think these Bd figures are not sufficiently suitable for publication because the samples have been wetted by long storage in a liquid medium, and during cross-section, their fragmentation occurred and may appear distorted. 

We thank the reviewer for the comment regarding the root data. We added Table S4 to describe these results. We also thank the reviewer for the comment regarding surface sterilization; this is performed automatically, and we have now added this information to the updated manuscript.

The second aspect that needs to be improved concerns the study of defense genes expression. The first point is that the authors seem not to use the same biological material to test endophytism (in roots) and defense gene expression (in leaves).

We thank the reviewer for his comment. As known from previous studies, endophytes induce systemic resistance. According to our results, after endophyte-inoculation of roots, information spread systemically throughout the plant, affecting the leaves' symptoms. However, we do not know which genes were expressed because the genes we examined were not affected by the endophyte as expected in the view of plant reaction to the pathogen. Other genes that we did not measure in our study were likely affected or a completely different mechanism involved in the induced resistance. However, the main objective of our work was to provide a suitable model for a deeper study of endophyte-mediated resistance and a tool for its reliable detection in host plant tissues, which we succeeded in doing, as we were the first to transfer M bolleyi to B distachyon and to design and thoroughly test primers for its accurate detection. We believe this opens up further avenues of study for the scientific community.

Indeed, they mention that Fc inoculations are performed “In the phase of the second offshoot in Bd and third leaf of wheat » which, albeit not being very precise, means rougly three-week old plants, whereas endophytism was tested 90 days after sowing that is much later. In this context, it is very difficult to compare expression analysis with the putative endophytic status of Mb as no result establishes that Mb is truly endophytic at the stage of inoculation by Fc.

Previous studies show that M. bolleyi is present in host tissues two weeks after seeding (e.g. Muray and Gadd 1981). The time of steady-state colonization and probably also the initial endophyte penetration into the host roots is essential in terms of resistance induction. The phenotypic response of different symptoms following infection with the pathogen confirms this. The main reason we evaluated endophyte's presence in roots later is that chlamydospores formation is at later stages, and we needed to be 100% certain that endophyte is indeed present to validate the usefulness of the inoculation method, the affinity of Bd to endophyte, and the newly developed PCR primers in this work properly. As shown in Table S4, all plants in the endophyte variants were colonized by Mb, and in all of them, chlamydospores were detected in root cells, which we consider being unquestionable proof of colonization that allowed us to verify the excellent function of the new primers.

Additional comments

Lines 240-256: the whole paragraph is unclear. The authors should first describe the induction of gene expression by Fc infection then describe how Mb presence may modify gene expression.

We thank the reviewer for this comment and we accepted the suggestion. We have reorganized the entire paragraph. In the new version of the manuscript, we described first the pathogen's effect, then the endophyte's effect, and finally, the effect of the endophyte on plants infected by the pathogen. In the new version of the article, we, therefore, split the original table. We are now presenting the assessment of the effect of endophyte on gene expression in hosts co-infected with the pathogen in a separate table, and we believe that the text is now much clearer.

Lines 242-243: BdChitinase1 and TaPR1.1 do show differential expression in the presence of Mb at 1 dpi. Why do the authors claim no statistical difference in gene expression

We thank the reviewer for this comment. We have added the information to the manuscript.

Minor revisions

Line 85: the 6 isolates were mixed together, detail in which proportion

We thank the reviewer for this comment. We have added the information to the manuscript.

Lines 234-235: gene names should be revised as used in the cited references (e.g. BdPR1 is BdPR1.1 in Kouzai et al. 2016)

We thank the reviewer for this comment. We have revised the name of PR1 gene. We used in our study primers for BdPR1-5, which Kouzai et al. 2016 recommended in their paper. It can look a bit like the range 1 to 5, so in the first version, we chose only the short version, which we found sufficient; however, it will indeed be better to keep the exact name. We corrected the name of the gene in the whole manuscript to correct form BdPR1-5.

---

## [Decision Letter · Decision Letter 1]

1 Mar 2022

Compatible interaction of Brachypodium distachyon and endophytic fungus Microdochium bolleyi

PONE-D-21-30855R1

Dear Dr. Matušinsky,

We’re pleased to inform you that your manuscript has been judged scientifically suitable for publication and will be formally accepted for publication once it meets all outstanding technical requirements.

Kind regards,

Sabrina Sarrocco

Academic Editor

PLOS ONE

Additional Editor Comments (optional):

Reviewers' comments:

Reviewer's Responses to Questions

**Comments to the Author**

1. If the authors have adequately addressed your comments raised in a previous round of review and you feel that this manuscript is now acceptable for publication, you may indicate that here to bypass the “Comments to the Author” section, enter your conflict of interest statement in the “Confidential to Editor” section, and submit your "Accept" recommendation.

Reviewer #1: All comments have been addressed

2. Is the manuscript technically sound, and do the data support the conclusions?

Reviewer #1: (No Response)

3. Has the statistical analysis been performed appropriately and rigorously? 

Reviewer #1: (No Response)

4. Have the authors made all data underlying the findings in their manuscript fully available?

Reviewer #1: (No Response)

5. Is the manuscript presented in an intelligible fashion and written in standard English?

Reviewer #1: (No Response)

6. Review Comments to the Author

Reviewer #1: (No Response)

7. PLOS authors have the option to publish the peer review history of their article (what does this mean?). If published, this will include your full peer review and any attached files.

Reviewer #1: No

---

## [Editor Report · Acceptance letter]

4 Mar 2022

PONE-D-21-30855R1 

Compatible interaction of *Brachypodium distachyon* and endophytic fungus *Microdochium bolleyi*

Dear Dr. Matušinsky:

I'm pleased to inform you that your manuscript has been deemed suitable for publication in PLOS ONE. Congratulations! Your manuscript is now with our production department. 

Kind regards, 

on behalf of

Dr Sabrina Sarrocco 

Academic Editor

PLOS ONE